# Follow-Up of Viral Parameters in FeLV- or FIV-Naturally Infected Cats Treated Orally with Low Doses of Human Interferon Alpha

**DOI:** 10.3390/v11090845

**Published:** 2019-09-11

**Authors:** Esperanza Gomez-Lucia, Victorio M. Collado, Guadalupe Miró, Sonsoles Martín, Laura Benítez, Ana Doménech

**Affiliations:** 1Department of Animal Health, Veterinary Faculty, Complutense University of Madrid, 28040 Madrid, Spain; 2Department of Animal Medicine and Surgery, Veterinary Faculty, Complutense University of Madrid, 28040 Madrid, Spain; 3Department of Genetics, Physiology and Microbiology, Faculty of Biology, Complutense University of Madrid, José Antonio Novais, 12, 28040 Madrid, Spain

**Keywords:** feline leukemia virus, FeLV, feline immunodeficiency virus, FIV, treatment, interferon, antigenemia, reverse transcriptase

## Abstract

Specific treatments for the long-life infections by feline leukemia virus (FeLV) and feline immunodeficiency virus (FIV) are either toxic, expensive or not too effective. Interferon α (IFN-α) is an immunomodulatory molecule which has been shown in vitro to decrease the release of infective particles. The aim of this study was to follow the progress of the clinical score and viral parameters of FeLV- and FIV-naturally infected privately owned cats treated with recombinant human IFN-α (rHuIFN-α, Roferon-A). Twenty-seven FeLV-infected cats (FeLV+) and 31 FIV-infected cats (FIV+) were enrolled in the study. Owners were instructed to orally administer 1 mL/day of 60 IU rHuIFN-α/mL in alternating weeks for four months. Blood samples were taken at the beginning of the study (M0), mid-treatment (M2), end of treatment (M4), and 6–10 months later (M10). Clinical status at these time points improved notably with rHuIFN-α treatment, regardless of the initial severity of the disease, an effect which lasted throughout the study in most animals (15 of the 16 FeLV+ symptomatic cats; 20 of the 22 FIV+ symptomatic cats) improved markedly their clinical situation. In FeLV+ cats plasma antigenemia (p27CA), reverse transcriptase (RT) activity, and proviral load decreased at M2 and M4 but increased again at M10 (“rebound effect”). The level of antigenemia or RT activity was below the detection limits in FIV+ cats, and the effect on proviral load was less marked than in FeLV+ cats. Taken together, these results indicate that rHuIFN-α is a good candidate for treating FeLV+ cats, but the “rebound effect” seen when treatment was discontinued suggests that additional studies should be conducted to clarify its effect on progression of the infection in cats.

## 1. Introduction

Feline leukemia and feline immunodeficiency are two of the most important viral diseases affecting domestic cats. Though both diseases are produced by the infection of retroviruses (FeLV and FIV, respectively) which are lifelong persistent, the pathogenesis of each disease is different. FIV infections have been described to progress following several stages similar to HIV infection in humans, leading to a final stage with a severe immunodeficiency that predisposes to many secondary infections and, eventually, to the death of the cat. This final stage of immunodeficiency is reached slowly, due to the progressive decrease of CD4+ T-cells, for which the virus has tropism. Naturally infected cats may be infected for many years without evident clinical signs if they live in an adequate environment (for example, in a household with appropriate care and food) [1], although recovery from FIV infection has never been documented. Even though there is a long clinical latency, there is some degree of FIV replication as antibodies are detectable early after infection and remain during all the life of the cat [2]. These antibodies are mostly against the main protein of the capsid, p24CA, and their detection constitutes the basis for serological diagnosis. However, some are triggered also by the envelope proteins (gp95SU and gp40TM), and their induction is used for a partially successful vaccine [3].

Feline leukemia is considered to be a severe disease that can progress to a fatal illness faster than FIV infection [4]. FeLV infection results in several different disease outcomes (reviewed in [5]). An unknown percentage of infected cats can control the infection in the initial stages and avoid the virus from spreading to target tissues (“abortive infection”). Many cats develop a “regressive infection”, in which the immune response partially contains the virus and viremia persists only for a few weeks. In some of these cats, the viremia is longer, and the virus infects haemopoietic stem cells in the bone marrow, where it can remain latent as proviral DNA integrated in the cellular genome. In approximately one third of all infected cats, the early immune response is unable to block the virus, and a massive replication and spread of the virus is produced, including the infection of hematopoietic stem cells in the bone marrow; this condition is called “progressive infection”, and cats become persistently viremic. Eventually the infection progresses to the development of related neoplastic (such as leukemia and lymphoma) and non-neoplastic (such as anemia, gingivostomatitis, and enteritis) diseases [4,6]. Usually, FeLV-infected (FeLV+) cats die within 1 to 3 years after the initial diagnosis due to all the pathologies and clinical complications the cats develop [5], mostly produced by secondary infections that may arise because of immunodeficiency. Because of the different outcomes, diagnosis of FeLV infection is usually complicated for clinicians. During the progressive infection, there is constant expression of viral proteins, especially of the capsid protein, p27CA, which may not assemble to form infective viral particles and the excess is secreted into the extracellular environment. Both infective and non-infective particles may be detected in FeLV infection: infective particles are complete virions which contain reverse transcriptase (RT), while non-infective particles are empty capsids [7]. Diagnosis is based on the detection of p27CA, as, unlike FIV-infected (FIV+) cats, antibodies are not as constant, due in part, to the excess of p27CA, which induces immune anergy which avoids the development of humoral response and which contributes to the persistence of the virus [8].

Both infections are difficult to treat because (i) antiviral drugs effective against these viruses are toxic or have significant side effects, (ii) retroviruses perpetuate in infected cells, and (iii) retroviruses produce a marked deterioration of the immune system, with severe complications and secondary infections, so the response to therapy in this phase is not as good as expected. For this reason, when cats are in this immunocompromised stage, the best option is to treat the clinical complications [9]. Several types of therapies have been tested in infected cats with different results: RT inhibitors (as nucleoside analogues, which are the most widely used), nucleotide synthesis inhibitors, receptor homologues/antagonists, integrase inhibitors, and interferons (reviewed in [9]).

The use of specific immune modulators, such as interferon, is a good option because they increase the innate immunity. This is suitable not only to decrease secondary infections which develop in retrovirosis, but also to decrease viral replication and spread to target cells and organs.

The use of interferons as antiretroviral therapy in cats has been addressed previously [9]. There are two types of interferon molecules with antiviral properties: type I (IFN-α; IFN-β and IFN-ω) produced by viral infected cells as first innate immune response against viral infection; and type II (IFN-γ), produced by T-lymphocytes and NK cells, which stimulates the cell-mediated acquired immunity against viruses. Type I IFN has been proposed for the treatment of FeLV- and FIV-infected cats based on previous results in HIV-infected patients [9,10]. Initial studies for treating retrovirus-infected cats used high doses of recombinant human IFN-α (rHuIFN-α, Roferon-A) administered orally. Though results were good, these high doses of human IFN stimulated the production of anti-IFN antibodies [11], and two different alternatives were proposed: prolonged administration of low doses of rHuIFN-α, and recombinant feline IFN-ω (rFeIFN-ω, Virbagen), which is the only IFN licensed by the European Medicines Agency (EMA) specifically for cats. As a molecule active in many cellular types of different species, rHuIFN-α has several advantages over rFeIFN-ω [12]. The antiviral activity of IFN-α is well-documented, and has been described to reduce in vitro the amount of viral infective particles of FeLV and FIV in feline cell cultures, without affecting the synthesis of viral proteins [13,14], though results in vivo are controversial [15]. In addition, rHuIFN-α is non-toxic, low-cost and easy to administer orally.

Several studies reported the effect of rHuIFN-α, mainly in FeLV-infected cats (reviewed in [9]). In general, when this cytokine is administered orally it has a local effect, stimulating the oral and pharyngeal lymphoid tissue, expanding into an immune cascade that finally has a systemic effect [9]. This has been associated with an improvement in the clinical signs, laboratory alterations and immunological markers such as the CD4/CD8 ratio of FIV+ or FeLV+ cats treated with IFN-α [10]. However, the effect of this immunomodulatory therapy on viral parameters of retrovirus-infected cats is not well documented. Contrarily, rFeIFN-ω has been shown to induce minor changes or no change in the CD4/CD8 ratio, proviral load, viremia and RT activity of infected cats, suggesting that the overall effect of IFN was on innate immunity [16,17,18]. Pedretti et al. [19] reported that no significant differences were observed regarding viral and proviral loads in a group of clinically sick FIV-infected cats throughout rHuIFN-α treatment.

The aim of the present study was to analyze the progress of viral parameters in FeLV- and FIV-naturally infected cats during and after treatment with rHuIFN-α, as information is lacking especially about whether changes in these parameters are long-lasting. This study reflects the everyday activity of the practitioner, in which when therapy is initiated, it is generally unknown when the animal became infected, and tries to reproduce the real conditions when treatment is administered by the cat owners at home.

## 2. Materials and Methods

### 2.1. Feline Patients and Administration of Treatment

Twenty-seven FeLV-infected cats (FeLV+) and 31 FIV-infected cats (FIV+) were enrolled in the study. All were privately owned, though most of them had been stray before. They had all tested positive to either virus by the serologic Snap Combo Plus (Idexx Laboratories Inc., USA) which detects FeLV p27CA and antibodies against FIV p24CA, and confirmed as positive by a nested PCR [20], and no discrepancy had been observed between both sets of results. Cats had been taken by their owners to one of four private veterinary clinics or to the Veterinary Clinical Hospital (Complutense University of Madrid), either because they had clinical signs suggestive of feline retrovirosis or for routine controls. Exclusion criteria included aggressive animals, pregnant and lactating females, or cats with severe renal, hepatic or cardiac diseases, with neoplasia or in the terminal stages of the disease. Many of the cats had been recently collected from the street or adopted from a feline community, and this was the first visit to the clinic. For this reason, the vaccination status was unknown in many cases. They were all naturally infected and the exact moment of the infection was not known; presumably, the cats were in different stages of the disease as the study included asymptomatic, with mild disease and with severe disease cats (see below).

Owners signed an informed consent form for their cat to be included in the study. They agreed not to treat the cat with any other immunomodulatory drug, but otherwise to care normally for the animal and not change their usual life habits during the study period (1 year). Animal handling, treatment, reagent manipulation and data collection were all carried out in compliance with guidelines for Good Clinical Practice, and Good Laboratory Practice of the Animal Welfare Committee of the Veterinary Clinical Hospital and the Complutense University, and the experimental procedures were approved by the Institutional Animal Care and Use Committee of the Complutense University (21 of June, 2007).

One mL doses containing 60 IU of commercial recombinant human interferon alpha 2a (rHuIFN-α2a, Roferon-A, Roche Diagnostics) were prepared aseptically by diluting the vials with sterile saline solution (0.9%), and stored at 4 °C until use. Each 1 mL dose was administered orally with the food or directly in the mouth for 7 consecutive days followed by 7 days of no treatment, alternating for four months, to a total of 56 doses in four months. Owners were trained as to how to administer treatment themselves at home, were responsible for administering all doses, and for following the instructions precisely.

### 2.2. Clinical Evaluation and Sampling

Owners were requested to take the cat to be examined at four different moments, or visits: M0, at the beginning of the treatment; M2, in mid treatment or month 2 (±15 days); M4, at the end of treatment or month 4 (±15 days); M10, four to eight months after finishing the treatment. Data collected at each visit (M2, M4, M10) were compared to the situation just before treatment was initiated (M0). The veterinarian in charge in one of the private veterinary practices associated with the study or in the Veterinary Clinical Hospital of the Veterinary Faculty of Madrid explored clinically and collected blood from the animals. At each visit, the clinical signs most frequently observed in FeLV and FIV infections (including loss of appetite, asthenia, altered mucosae, or lymphadenopathy [4]) were rated 0, 1 or 2 [4] according to their relevance in the disease, and their severity was evaluated in cats to obtain a clinical score (CS) up to a maximum score of 15 [4]. According to this CS at M0, cats were classified into three clinical groups (CG): CG1, with no clinical signs (asymptomatic); CG2, with a CS 1–5 (mild disease); CG3, with a CS ≥ 6 (severe disease).

Blood samples (2 mL) were collected from the cephalic or jugular veins and distributed into a tube with EDTA and another with heparin-lithium and sent immediately to the Department of Animal Health of the Veterinary Faculty of Madrid.

### 2.3. Detection of FeLV p27CA, FIV p24CA and Retrotranscriptase Activity (RT)

The antigenemia was determined by evaluating the concentration of the capsid proteins FeLV-p27CA and FIV-p24CA in the plasma of the cats diluted 1:2 in PBS, using the commercial tests PetChek FeLV Antigen Test and PetChek FIV Antigen Test (Idexx), respectively. The amount of infective viral particles was evaluated by the RT activity of FeLV and FIV as measured by two tests of Cavidi Tech (Uppsala, Sweden): C-type-RT^TM^ Activity Assay, and Lenti-RT^TM^ Activity Assay, respectively for FeLV and FIV. Appropriate negative controls and the positive controls included in the kits were used. Color development was measured in a spectrophotometer (Tecan Spectra-Fluor A-5082, Crailsheim, Germany), and data were interpreted following the manufacturers’ directions.

### 2.4. Evaluation of FeLV and FIV Proviral load by Real Time PCR (rtPCR)

Genomic DNA was extracted from whole heparinized blood using DNeasy Tissue Kit (Qiagen), following the instructions of the manufacturer, and it was stored at −20 °C until used. Proviral DNA was quantified by real-time PCR using previously reported primers and fluorescent probes (0.2 µM) (Cy5 for FeLV, FAM for FIV) [21,22] in a thermocycler Mx3000P (Agilent Stratagene, USA). Platinum Taq DNA polymerase (1U/µL) and other PCR reagents were purchased from Biotools, except for the primers and the fluorescent probes (Genosys-Sigma).

The results of real-time PCR were normalized to cellular glyceraldehyde-3-phosphate dehydrogenase (GAPDH) using primers GAPDHfor (5´-CGGAGTCAACGGATTTGGTCGTAT-3´) and GAPDHrev (5´-**ACTGAACCTGACCGTACA**AGCCTTCTCCATGGTGGTGAAGAC-3´) [23] which amplify a 307 nucleotide (nt) fragment. GAPDHrev included in its sequence 18 nucleotides (underlined bold letters), called Z sequence, which is incorporated to the nascent DNA strand, and emits fluorescence when it combines with Amplifluor™ UniPrimer™ (Merck Millipore). The instructions of the manufacturer were followed, using 0.5 µM of GAPDHfor and 0.05 µM of GAPDHrev and 1 U/reaction of Titanium Taq DNA polymerase^®^ (Biotools). Results were expressed as the ratio proviral DNA Ct/total GAPDH Ct. A variation >20% with respect to the value at M0 was considered as decrease or increase in the viral load.

The amplifying programs were similar to those described by Leutenegger et al. [21] and Pinches et al. [22] with slight modifications to allow the joint amplification of proviral genomes and GAPDH, introducing an elongation step necessary for the UniPrimer system (3 min at 72 °C for FeLV rtPCR and 1 min at 72 °C for FIV rtPCR). Fluorescence was measured with a FAM filter.

### 2.5. Statistical Analysis

Results were statistically analyzed by the Informatics Service of the Data Processing Center of the UCM using Origin Pro 7.5 (https://data.library.virginia.edu/research-software/originpro/), SPSS (IBM Corp. Released 2015. IBM SPSS Statistics for Windows, Version 23.0. Armonk, NY, USA) and Statgraphics Centurion XVIII Windows (www.statgraphics.com). Differences < 0.05 in the Student’s *t*-test (real-time PCR) and Wilcoxon test (CS, p27 and RT activity) were considered significant. Only significant differences are mentioned in the text.

## 3. Results

The study involved naturally infected cats with different clinical status, possibly reflecting diverse stages of the disease. According to the CS when they enrolled the study 11 (40.7%) and 9 (29.0%) of the FeLV+ and FIV+ cats, respectively, were clinically healthy (CS = 0) and were ascribed to CG1; 8 FeLV+ (29.6%) and 13 FIV+ (41.9%) cats were considered to have mild disease as their CS was 1–5 and were ascribed to CG2; the remaining 8 (29.6%) of the FeLV+ cats and 9 (29.0%) of the FIV+ cats had a CS ≥ 6 and were considered to have severe disease (CG3). The clinical situation of these cats at M0 has been published before [4]. Eleven of the FeLV+ and 20 of the FIV+ cats were males. Two thirds of the FeLV+ (18/27) and FIV+ (21/31) populations were neutered or spayed. Their age ranged between 6 months and 14 years, with FeLV+ being considerably younger (0.3 to 6 years). Most cats were mixed breed (commonly known as domestic shorthair), while only 7 of the 58 cats were of a breed (4 Persian, 3 Siamese). Though owners had been instructed to bring their cat to all three visits, for different reasons several failed in this obligation and some data are missing. Only cats which had been brought to at least two of the three visits were included in the study. Most owners took their cats while treatment lasted (high fidelity during M2 and M4), but only 44.4% of the FeLV+ and 77.4% of the FIV+ cats were taken to M10.

During the study period, most of the cats (15 of the 16 FeLV+ symptomatic cats; 20 of the 22 FIV+ symptomatic cats) improved their clinical situation markedly, regardless of their initial clinical status (Table 1). None of the cats of CG1 (healthy) developed clinical signs during the study. FeLV+ cats took longer to improve than FIV+ cats, as only 46.6% of them were seen to improve at M2 but 100% had a better CS at M10 than at M0, while 84.2% of the FIV+ cats had already improved their CS at M2. Improvement associated with treatment with rHuIFN-α was seen to be statistically significant (Wilcoxon test) at all three visits: mid-treatment (*p* < 0.001), end of treatment (*p* < 0.005) and end of treatment (*p* < 0.005). Two FeLV+ and one FIV+ cats died during the study, all of them of CG3. Improvement was especially noticeable in FeLV+ cats in CG3, as they passed from an average CS of 7.62 at M0, to 6.0 at M2, 3 at M4 and 0 at M10. The clinical signs which improved the most and even became unnoticeable in most cats were loss of appetite, asthenia, weight loss and respiratory alterations. Lymphadenomegaly and oral lesions took longer to resolve.

### 3.1. Plasma Antigenemia

The capsid protein of FIV, p24CA, was not detected in any of the sera of cats. Contrarily, FeLV-p27CA, was detected in all the FeLV+ cats. When compared to M0, the concentration of p27CA decreased in 60.0% of the cats at M2, and in 59.1% at M4, but increased in 32.0% and 27.3% of FeLV+ cats at M2 and M4, respectively. However, at M10 66.7% of these animals had a higher concentration of p27CA than at M0 (Figure 1A). In addition, when compared to M0 the average concentration of FeLV-p27CA decreased 18.1% and 23.6% at M2 and M4, respectively, increasing 31.5% at M10 (Figure 2A). Thus, a “rebound pattern” was observed in interferon-treated FeLV+ cats. Surprisingly, the highest average value of p27CA at M0 was in the group of healthy cats (CG1), but the initial values of p27CA were similar in all CG (0.46−0.60 mg/µL). However, the percentage of cats in which FeLV-p27CA was reduced at M2 when compared to M0 was higher in CG1 and CG2 (70.0% and 75.0%, respectively), than in CG3 (28.6%) (*p* < 0.05).

### 3.2. Reverse Transcriptase (RT) Activity

As with the capsid protein, RT activity was not detected in FIV+ cats. On the other hand, the RT activity was detectable in 66.6%, 28.0%, 31.8%, and 58.3% of the treated FeLV+ cats at M0, M2, M4 and M10, respectively. Around three fourths of the cats that had a positive RT activity value at M0 had a decreased reading at M2 and M4, but a much smaller percentage had an improved RT activity value at M10 (Figure 1C). In addition, the detection of RT activity increased progressively in cats in which it was initially undetectable (Figure 1D). The highest improvement on the RT activity of treated FeLV+ cats was observed at M2, when the highest percentage of these cats had undetectable levels of this parameter. However, the concentration of RT activity was the lowest at M4, over 95% lower than at M0, increasing when treatment was discontinued (Figure 2B). All these results support the “rebound pattern” mentioned above. No significant differences were observed in the progress of RT activity in relation to clinical status at M0.

### 3.3. Proviral Load

The proviral load was quantified in the peripheral blood of FeLV+ and FIV+ cats using real time PCR to determine how it varied with treatment. As explained in the Material and Methods section, values were considered to increase or decrease when the percentage of provirus/GAPDH varied by more than 20% with respect to the value at M0.

As with the other tests, in FeLV+ cats there was an improvement of proviral load at M2 and M4 when compared to M0 (40.0% and 59.1% decrease, respectively), but a regression at M10 (33.3% decrease compared to M0) (Figure 1B). The percentage of cats that increased their proviral load remained stable throughout the study (16.0% at M2, 13.6% at M4 and 16.7% M10) (Figure 1B). In addition, the lowest average proviral load was at M4, 19.8% lower than at M0. There was a positive correlation between the concentration of FeLV-p27CA protein and proviral load (*p* < 0.05). No significant differences were observed in the evolution of the proviral load in the different clinical groups, but female cats had increased FeLV proviral loads between M0 and M4, while male cats did not (*p* < 0.05).

The effect of rHuIFN-α on the proviral load was notably less in FIV+ cats than on FeLV+ cats (Figure 1E). Proviral load decreased in 29.2% of the FIV+ cats at M2 (vs. 40.0% of the FeLV+ cats at M2). At the following visits, the percentage of cats with decreased proviral load was less than at M2 (15.4% and 8.3% at M4 and M10, respectively). On the other hand, the percentage of cats in which proviral load increased was also lower than the corresponding values in FeLV+ cats (no cats in M2, one in M4 and two at M8).

### 3.4. Combined Results of the Viral Parameters in FeLV+ Cats

Results obtained with cats included in the same clinical group were compared in order to better understand the viral progression depending on the clinical status (Table 1). Few differences were observed throughout the study in the concentration of FeLV-p27CA, RT activity, or proviral load between the three different clinical groups, though the starting RT activity was highest in cats of CG3 (severe disease), and improved the most in this group.

With regard to the progress of the viral parameters studied, the highest percentage of cats which improved their values was between M2 and M4. M4 was the visit in which the average FeLV-p27CA, RT and proviral load was the most reduced when compared to M0 (23.6%, 96.2% and 19.8%, respectively) which seems to indicate that it corresponds to the highest response to treatment (Table 1). At M10 the number of animals with a favorable progress was drastically reduced.

## 4. Discussion

This study was conducted to analyze the effect of oral treatment with low doses of rHuIFN-α (Roferon-A) on the viral parameters of domestic cats naturally infected with FeLV or FIV, since from our previous experience, we had observed that the clinical status of infected cats improved notably with IFN treatment. The results show that treatment with low doses of rHuIFN-α was associated with an improvement of the viral parameters studied in FeLV+ cats as long as the animals were treated, but reverted to levels similar to the original ones within 4–8 months upon treatment discontinuation (“rebound effect”). Contrarily, in most animals, the clinical status improved steadily throughout the one-year study.

An important aspect of this study is that it represents what practitioners find at the clinic. They do not usually know when the cat was infected, the stage of the disease, the immune status of the animal, the CD4/CD8 ratio or the proviral load. Cats may have one or more clinical signs or be totally asymptomatic. Treatment with IFN may produce unexpected results, ranging from no visible effects to an apparent clinical improvement. A big concern for the veterinary practitioner is whether clinical improvement correlates with a true improvement of the infection, i.e., a decrease in viremia or the proviral load of the cat, or whether the infection may revert when IFN treatment is discontinued. In this type of study, average values are not easy to use, as they do not provide a true measure of the progress in each cat. Thus, a single-arm clinical study was applied, similarly to other previously published studies with naturally infected cats [17,18,24]. We agree with Gil et al. [17], who suggest that values at D0 are potentially more representative than a placebo in studies in field conditions, in which animals have been infected at unknown moments. Therefore, each cat at day 0 was considered its best untreated control, and the response to treatment was evaluated over time by comparing each parameter in each individual cat at each visit with their situation before beginning treatment (M0). This eliminated the need to use a placebo-treated or non-treated group of cats, which would have been ethically controversial, as owners demand a solution for their cats [17,18,24]. The control group would ultimately be necessary if we were studying experimentally infected animals, where all cats should be in the same stage of infection with the same viral strain. In our study, we had a heterogeneous group of cats and it was impossible to know the point and stage of the infection in each animal.

Capsid protein or RT activities were easily measured in the plasma of FeLV+ cats. FeLV-p27CA may be released by infected cells not associated with viral particles and is thus used as a marker of antigenemia [13,25,26], while RT is present only inside virions and is proportional to the number of infective viral particles [7]. Thus, the follow-up of both proteins provides information on the progress of FeLV in infected cats treated with rHuIFN-α.

Around 60–75% of FeLV+ cats had improved FeLV-p27CA and FeLV-RT activity during M2 and M4 of treatment. This suggests that rHuIFN-α decreases antigenemia and circulating infective particles. Similar results with respect to the concentration of p27CA in blood had been obtained previously [11], and also agree with previous data which suggest that p27CA level is an indicator for the outcome of the FeLV-infection and may differentiate between regressive and progressive infections [27]. According to our results, rHuIFN-α had a more evident effect on RT activity than on p27CA, as the average value of RT activity decreased more and in a higher percentage of animals than that of p27CA. This supports the observation that interferon acts on the assembly and formation of viral particles more than on the expression of viral proteins, as was derived from previous in vitro studies [13,28]. Affecting this step and not the protein synthesis, p27CA may be released, explaining why the plasma concentration of p27CA decreased less than the RT activity.

The presence of high levels of p27CA in FeLV+ cats is associated with the final progressive stages of this infection [25], and in fact, this parameter has been used to follow the progress of the infection upon treatment [27]. In agreement with the European Pharmacopoeia, a cat is designated as progressively infected when FeLV-p27CA is positive for three consecutive weeks [27]. Thus, low p27CA concentration in plasma may mean a positive effect of rHuIFN-α treatment, similarly to what happens in HIV patients, in which the decrease of circulating p24CA levels induced by IFN-α correlates with the increase of CD4+ T-cells and an antitumoral response against Kaposi’s sarcoma [29]. However, it is important to mention that in our study, rHuIFN-α drastically improved the clinical score of all animals, which did not result in a rebound effect during the period that the study lasted. Thus, in our study there was no correlation between the clinical and the viral status after treatment was discontinued. In this sense, our results agree with observations by others who indicated a lack of correlation between viremia and clinical scores in naturally FIV-infected cats [18].

The study of the proviral load in retroviral infections provides information about the number of infected cells, and thus, the progress of the infection and also the ability of the cat to transmit the infection. In the case of FeLV+ cats treated with rHuIFN-α, our results show that at M4 59.1% of them had a decreased proviral load as compared to M0. However, the percentage decreased to 33.3% at M10, suggesting that, as with other parameters, the effect of rHuIFN-α ceased when treatment was discontinued. In general, the effect of rHuIFN-α on proviral load was not as positive as on RT activity or p27CA protein. All these results are slightly better than those reported for rFeIFN-ω (Virbagen), as no substantial improvement of the proviral load had been observed in FeLV+ or FIV+ Virbagen-treated cats [16], or even the proviral load had increased in FIV+ cats after subcutaneous administration of this molecule [18].

The concentration of p27CA and RT activity and proviral load in plasma experimented a “rebound pattern” at M10 in FeLV+ cats. These results seem to indicate that the effect of rHuIFN-α was reduced after discontinuing treatment, and agree with our observations on biopathological parameters in these cats (manuscript in preparation). Zeidner et al. [11] described the decrease of the concentration of p27CA at the beginning of treatment with rHuIFN-α, followed by its increase at the end, similarly to our results. However, these authors detected the presence of anti-rHuIFN-α specific antibodies, which could have blocked its effect. A rebound effect in viremia was also observed after a short-term treatment with an integrase inhibitor drug in FeLV infected cats [27]. According to our results, the viral situation of FeLV+ cats was worse in 16% to 67% at M10 than at M0, even in those infected cats which initially had undetectable levels of RT activity. This could mean that upon the discontinuation of treatment, FeLV+ cats could experience a worsening of the viral parameters, usually along with anemia, and life expectancy could be short. The possibility of a longer treatment with rHuIFN-α, not explored so far to the best of our knowledge, should be addressed in future studies.

Treatment with rHuIFN-α had a much lesser effect on FIV-infected cats. Though both FIV-p24CA and FIV-RT activity could be detected in in vitro studies [14], no FIV p24CA or RT activity was detected in the plasma of FIV+ cats. Several reasons could account for this. First, the virus may be latent, and not be expressed to a high degree [30,31]. However, around two thirds of the animals had clinical signs at the beginning of the study, which would indicate an active infection. A second possibility could be that circulating antibodies against FIV, which are undoubtedly present as they are detected by the Idexx device, could block free virus in blood [32], though this possibility would not explain the absence of RT activity in the blood samples. However, the most likely explanation is that cats were not in the short antigenemia window, as FIV is mostly inside cells and antigenemia or presence of viral particles in plasma is transient and not maintained [33]. Other authors have pointed out the difficulty in detecting viremia in cats naturally infected with FIV with an unknown duration of infection [18].

The proviral load of a very high percentage (70–83%) of FIV+ cats treated with rHuIFN-α was not modified and values were similar to those at M0, suggesting that treatment does not evidently improve this parameter. Contrarily to our results, a study by Pedretti et al. [19] evidenced the decrease in the proviral load in sick FIV+ cats (possibly similar to CG3 cats of our study), both in placebo- and in rHuIFN-α-treated cats, and no significant differences were observed between both groups.

Previously, it has been proposed that direct antiviral effects were unlikely after oral administration of rHuIFN-α, and that its action could be related to an indirect effect through the stimulation of local lymphoid tissue in the oral cavity that triggers an immune response against viruses (reviewed in [9]). However, the results presented in this paper suggest that oral rHuIFN-α does seem to have an effect on the viral parameters studied, also corroborating in vitro studies published by our group [13]. A possible drawback of our study could be the production of antibodies against human IFN, which has been demonstrated to happen by treating with high doses of this cytokine [11]. However, at the doses and administration route used in this study (60 IU rHuIFN-α/day/cat orally) we did not detect the presence of specific antibodies at any of the samples collected (results not shown).

With respect to other antiviral treatments, rHuIFN-α has the advantage of having no side effects or toxicity and not developing drug-resistant strains of virus, as described in treatments with antiviral drugs (such as AZT, which induces non-regenerative anemia, reviewed in [9,34]). Other advantages are the easy availability and low cost of this cytokine, and the possibility of per os administration. In addition, most antivirals used in cats are less effective against FeLV, as these drugs are specifically targeted for HIV in humans, and are more related to FIV (and their enzymes have similar sensitivities to inhibitors). There are several data that show a real effect of antivirals in infected cats, and in many of these compounds and similar to our results, the positive effect disappears after the treatment is stopped (reviewed in [9]).

The study had several limitations. One of these was including cats at different stages of the infection. Another was that the results were dependent on the adherence of the cat owners to the treatment administration and taking the cat to the clinician at the appointed times. As owners possibly judge the progress of the infection by the presence or absence of clinical signs that they can observe, some may have relaxed in their duty of administering some of the doses of treatment upon considering that the cat had improved clinically. Additionally, M10 was the visit at which the fewest cats were taken to the veterinary practice, which could be due to owners’ negligence, a long-lasting improvement, or dramatic impoverishment of the clinical situation of the cat. As a whole, these events which depended on the owner were impossible to control. Nevertheless, these limitations do not decrease the relevance of the results presented and, as mentioned above, they more accurately represent the situation encountered by the clinician at the everyday practice. Lastly, the fact that PCR to determine the proviral load was done on heparinized blood, instead of using blood conserved in EDTA or sodium citrate, may have decreased the values obtained, as heparin may block Taq polymerase [35]. However, the use of normalized values compared to GAPDH, a housekeeping gene, may have diminished the importance of this limitation; in addition, previously we had compared results using blood collected on heparin or on EDTA tubes and found no measurable variations.

## 5. Conclusions

When used in FeLV- or FIV-naturally infected cats, oral treatment with rHuIFN-α may cause an improvement in the antigenemia (p27CA) and RT activity and proviral load in FeLV+ cats, while it possibly affects viral parameters in FIV+ cats to a lesser extent. Taking into account the results in the viral parameters and the clinical improvement, the oral treatment with rHuIFN-α could be an interesting alternative for the therapy of cats infected with FeLV and FIV, mainly due to its low toxicity, the low cost of these compound, and the ease of its oral administration compared to the injectable protocol for other compounds, which facilitates its supply by the owner at home. The reversion to initial values of several of the factors studied after treatment was discontinued suggests that treatment with rHuIFN-α had not been as long as necessary, or maybe that cycles should be repeated to consolidate the improvement. More studies are necessary to determine whether the rebound pattern is a natural progress of the infection, or whether it may be mitigated by longer treatment.

## Figures and Tables

**Figure 1 viruses-11-00845-f001:**
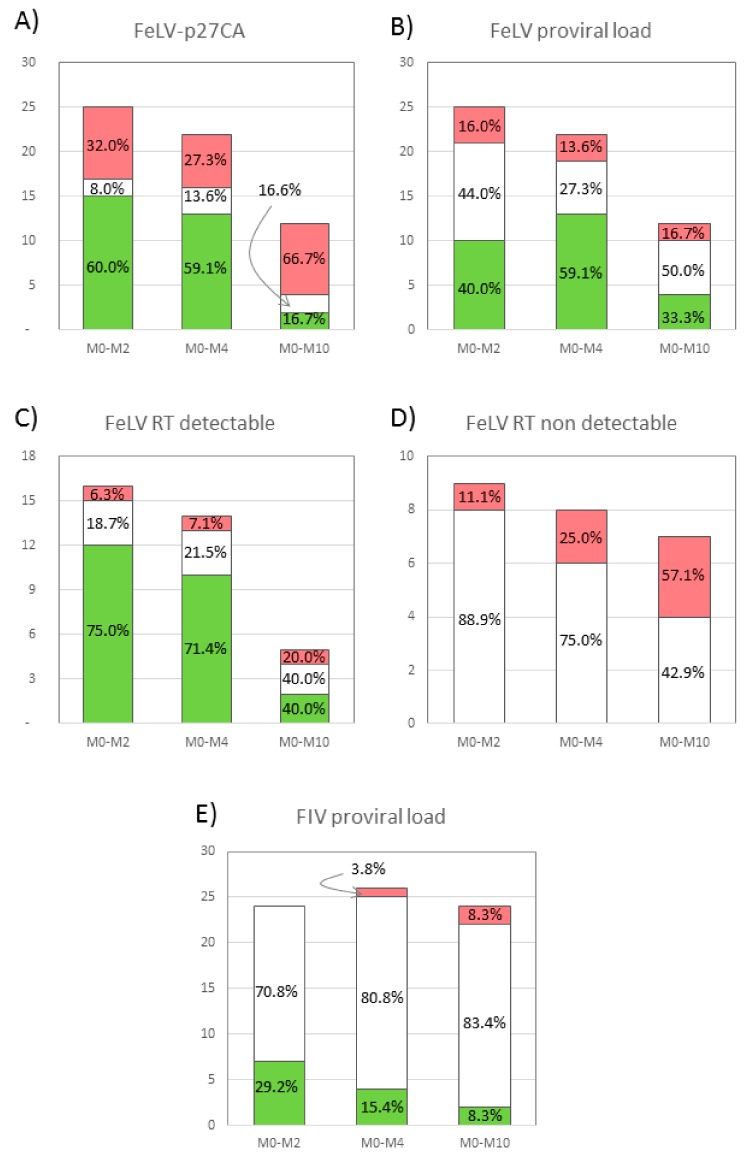
Progress in peripheral blood of rHuIFN-α-treated cats of FeLV-p27CA (**A**), proviral load in FeLV+ cats (**B**), FeLV-RT in cats with detectable levels of this parameter at M0 (**C**) and undetectable levels at M0 (**D**), and proviral load in FIV+ cats (**E**). Each column represents the number of cats (including the percentage) in which the parameter studied at M2, M4 or M10 was >20% better (green) or worse (red) than the respective value at M0. White sections represent the number of cats in which the value was <20% better or worse than that detected at M0.

**Figure 2 viruses-11-00845-f002:**
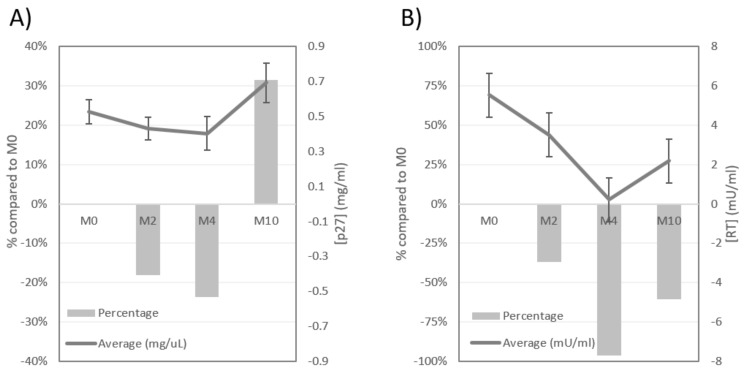
Progress of FeLV-p27CA (**A**) and RT activity (**B**) in plasma of FeLV+ cats treated with rHuIFN-α. Columns show how much the average concentration of p27CA or RT activity had increased or decreased as compared to the average at M0; line indicates the average concentration of p27CA (mg/µL) or RT activity (mU/mL) at each time point; bars in line indicate standard error. Note the favorable progress of cats at M2 and M4, and the unfavorable progress at M10.

**Table 1 viruses-11-00845-t001:** Progress of the clinical status and of the average of the viral parameters analyzed in FeLV+ cats belonging to clinical group (CG) 1 (asymptomatic), CG2 (mild disease), and CG3 (severe disease) at the different time points. CS, average clinical score. p27CA, average concentration of FeLV-p27CA (mg/μL). RT, average RT activity (mU/mL). Proviral load, ratio FeLV Ct: GAPDH Ct. Numbers in parenthesis are the standard error.

	CS	p27CA	RT	Proviral Load
CG1				
M0	0.00 (0.00)	0.60 (0.13)	0.86 (0.52)	0.72 (0.04)
M2	0.00 (0.00)	0.49 (0.08)	0.11 (0.09)	0.78 (0.05)
M4	0.00 (0.00)	0.55 (0.18)	0.12 (0.12)	0.88 (0.10)
M10	0.00 (0.00)	0.63 (0.16)	0.65 (0.39)	0.66 (0.06)
CG2				
M0	3.50 (0.05)	0.46 (0.13)	1.22 (0.59)	0.61 (0.06)
M2	3.25 (0.84)	0.22 (0.07)	0.28 (0.28)	0.72 (0.06)
M4	1.75 (0.62)	0.40 (0.17)	0.62 (0.40)	0.70 (0.08)
M10	2.00 (0.00)	0.85 (0.19)	3.29 (3.29)	0.77 (0.04)
CG3				
M0	8.00 (0.84)	0.49 (0.08)	3.13 (1.46)	0.73 (0.06)
M2	6.17 (1.48)	0.58 (0.16)	2.51 (1.55)	0.81 (0.07)
M4	3.00 (1.91)	0.20 (0.10)	0.25 (0.15)	0.88 (0.04)
M10	0.00 (0.00)	0.66 (0.40)	0.40 (0.37)	0.69 (0.03)

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
