# Peer review of "Follow-Up of Viral Parameters in FeLV- or FIV-Naturally Infected Cats Treated Orally with Low Doses of Human Interferon Alpha"

_viruses, 2019, doi:10.3390/v11090845_

Round 1
Reviewer 1 Report
The paper describes the evolution of the clinical score and viral parameters in 27 FeLV- and 31 FIV-naturally infected cats treated with recombinant human IFN-α and found an improvement in the antigenemia and RT activity and proviral load in FeLV+ cats, and to a lesser extent in FIV+ cats.
Overall a well written paper that add important information on the treatment of feline retrovirosis, in particular on the effect of human IFN-α on FeLV parameters.
The major flaw in this manuscript is on the lack evaluation of clinical score and clinical signs of retrovirus infected cats enclosed in the study. In my opinion the study will be more complete and clear if the clinical aspect of the cats, and also hematological and biochemical data if available, will be enclosed in this manuscript. For example CD4 and Cd8 evolution evaluation could better clarify the effect of human IFN-α in FIV cats.
If the authors do not agree to discuss in this paper also the other aspects they have evaluated in these cats, I suggest to eliminate the clinical score evaluation from this paper and discuss and try conclusion only from viral parameters, as indicated in the title of the paper.
Other specific comments as follow
Abstract
Line 21: “Owners were instructed to administer orally 60 IU/mL/day in alternating weeks for four months.” Not clear what is the dose administered orally, if not specified that a cat was administered 1 ml/day
Line 27: “Antigenemia or RT could not be measured in FIV+-cats”, as is written seem that th ewproblems is with the methods to evaluate these parameters in FIV+ cats
Line 29-30: “but the “rebound effect” seen when treatment was discontinued is preoccupying.” The term “preoccupying” is somewhat inusual, I suggest to write that additional studies should be performed on this effect to clarify the effect on progression of the infection in the cat
Introduction
Line 90: “Roferon-A has several advantages over Virbagen” here and throughout the manuscript I suggest to use the term human IFN-α or rHuIFN-α and recombinant feline IFN-ω instead of Roferon-A and Virbagen
In addition It should be better for the reader have the advantages listed here instead of in the final discussion
Line 111-112: “This study reflects the everyday activity of the practitioner, in which the viral status of animals is generally unknown when therapy is initiated, and tries to reproduce the real conditions when treatment is administered by the cat owners at home”. it is not clear what means this sentence in this point of the manuscript, I suggest to move it in the discussion of the limits of the study
Material and Methods
Line 115: how the cats were enrolled in the study? For what reasons cats were tested for retrovirus? How is the vaccination status of these cats? I feel that we need more information on the cats enrolled in the study
Line 118: do you think that mixed breed is equivalent of domestic shorthair (DSH) or are different cats?
Line 120: both FeLV and FIV results were PCR confirmed? Please clarify
Line 126: what does it mean “administer the treatment completely”?
Line 133: see the above comment on the dose in the abstract
Line 142: “M10, four to eight months after finishing the treatment”. Why a so broad time interval for the same control? Is this study prospective or retrospective in its nature?
Line 154-162: are the tests used in this section validated in the cats? In particular why the plasma sample were diluted 1:2 with PBS?
Line 178179: “A variation >20% with respect to the value at M0 was considered as decrease or increase in the viral load”. How was chosen the cutoff of >20% to identify a decrease or increase in viral load?
Line 186-187: Could you explain the utility of ANOVA and Chi square in the statistical analysis?
Results
Line 195-198 and line 116-118: I suggest to move all demographic data of the cats enrolled in the study here in the results section instead to have some in the M&M and some here. In addition it would be interesting to know to which breed (line 198) belong the infected cats
Line 201: “(Table 2 and manuscript in preparation)”. How can the reader could understand the results presented in this manuscript if some of these belong to another manuscript?
Line 205 and 206: the parameters evaluated decrease and increase in a statistically significant way? If yes could you provide the P value? In addition I suggest to write the exact P value instead of a generic P<0.05, so the readers could understand the real magnitude of the difference.
Figure 1: I have some problem to read the small numbers inside the pie. Should these results summarized in a table instead of with as such number of small pies?
Table 2: I suggest to delete the Clinical score column as we have no sufficient information on how this score was calculated
Discussion
Line 291-292: The clinical status of infected cats improved notably with IFN treatment (manuscript in preparation).” Again for improve clarity I suggest to enclose all the data in this manuscript or do not cited the not published results at all
Line 313-314: “It is important to mention that both FIV p24CA and RT could be detected in in Vitro studies”. I suggest to move this sentence at the start of this paragraph
I suggest to add a paragraph in the discussion relative to the limits of the study
Author Response
Please, see attached file.

Reviewer 2 Report
The authors describe viral parameters of cats naturally infected with FIV or FeLV and treated orally with low doses of human interferon alpha. The aim of the study in the abstract indicates that also clinical parameters are investigated; however, these results are not presented in detail. The manuscript presents the viral parameters of FeLV-infected cats and proviral loads of FIV-infected cats under therapy (single-arm study). The results are interesting: a decrease of the FeLV capsid antigen was found, followed by a rebound effect after the therapy was stopped. These data should be published. However, the manuscript needs major revision. Some sections are difficult to understand, possibly due to language issues. The results should be streamlined, and statistics revised. The clinical data should be added to the manuscript to present a complete story.
Title and throughout the text: “evolution” implies – at least in biological terms – that the virus changed the characteristics. This was not investigated. Please delete or replace this term.
Abstract: the authors should substantiate their statements by adding numbers, percentages, significances. Examples:
- line 17: “in vitro … shedding of infectious particles” – unclear what is meant here. Virus is usually shed by animals; thus, in vivo. Please rephrase, since here in concerns in vitro.
- line 20: “belonging to households” – privately owned cats?
- line 25: “most animals”. Can the authors give a number or a percentage?
- line 26: was this a significant decrease?
Introduction
- Language could be improved. E.g. line 37 “long-life persistent”; line 19: FeLV is not a disease; it’s a virus.
- FeLV pathogenesis: The authors do not use the currently used terminology for the different courses of FeLV infection; this needs to be adapted to the recent literature. I suggest using the following terms: “progressive infection” and “regressive infection”. Line 53/53: a cat that is progressively infected does not have latent infection. This sentence is contradictory. Line 54: during which infection? Probably progressive infection is meant here. Line 58: “persistent viremia”, probably progressive infection is meant. Line 60/61: probably progressive infection is meant here? Is reference 4 the right reference for this statement (lifespan 1-3 years in cats with progressive infection)?
Aim of the study: How is this aim is different from that of previous studies and how does it add new information to the field? Also see: “the use of interferon as antiretroviral therapy in cats has been extensively studied” (line 79).
Lines 86-87: High doses of human interferon-alpha induced anti-IFN antibodies. Could the authors please add a reference to this sentence.
Lines 89/90: I am not quite convinced about this general statement that human interferon alpha has advantages over the feline interferon omega. E.g. compare the FeLV guideline by ABCD, table 3, summary of immunomodulatory substances against FeLV.
M&M
Lines 146-148. Could the authors please list “the clinical signs most frequently observed in feline retrovirosis”? did they use a clinical score sheet? If so, please add to the publication. Please delete the term “feline retrovirosis”.
Please do NOT use rtPCR for real-time PCR. This is confusing, since RT-PCR is used for reverse transcriptase PCR. Please list this parameter (real-time PCR result) in the results section as “proviral load”. Use “RT activity” instead of only “RT” in the results section (for RT activity).
Why did the authors used heparinized blood for PCR? Heparin can inhibit polymerase activity. EDTA or sodium citrate would be more adequate. Maybe add to limitations of study in the discussion?
Why did the authors use a human specific GAPDH assay? There are feline assays published. Did the authors make a sequence comparison or is there published data on the specificity of the assay for feline genomic DNA? Maybe add to limitations of study in the discussion?
Data analyses: Line 186: why did the authors use parametric tests (ANOVA) – was the data normally distributed? Line 199: statistics: “some data are missing”. It should be specified, which data was missing/available for statistical analysis.
Results
Lines 190 ff: the clinical scores cannot be judged like that. The conclusion that during the study period most of the cats improved clinically is not supported by data and statistics. Please add detailed results.
Figure 1: the content of the circles, particularly the colored sections, is hard to read. There seems to be a mislabeling (twice FeLV RT)
Figure 2: only present the mg/ml in Fig. 2; consider using box plot or similar, since probably the data is not normally distributed. The percentages are suggestive of a larger change than present; delete from figure (may be kept in the text).
Minor: The tables should be mentioned in the text in the correct order. Now Table 2 is mentioned in line 201, then table 1 in line 231.
RT results and provirus results: I suggest presenting these results also in a figure (like Fig. 2), where the reader can assess the actual data at each time point and judge by him/herself the de-/increase of the activity over time. Table 1 seems not helpful.
Provirus results (line 250): is there any justification to consider a change of 20% to be biological relevant?
Please delete chapter 2.3. and table 2. Redundant.
Discussion
Why not start with the discussion of the most relevant results? Transient decrease in FeLV viral loads (lines 312 ff). Please discuss, why the decrease was only transient. Could the therapy be continued or restarted? What about development of anti-human IFN alpha antibodies? Could this be measured (at least discuss)?
Move the low toxicity up (as a positive statement; lines 376 ff), plus low-cost, plus p.o. administration.
Move FIV detection further down (since mainly negative)
Line 316 – sentence does not make sense. FeLV p27 IS circulating antigen (“is a measure of circulating antigen”?)
Please do not mix FIV and FeLV in the discussion section.
Lines 366 ff: not sure I understand: “upon the discontinuation of treatment, FeLV-positive cats could develop persistent viremia”. Did you not include only cats with progressive infection (persistent viremia)? If this was not the case, please add to discussion (also cats with regressive infection could have been included, since the cats were tested only at one timepoint for antigenemia prior to the start of the study? Clear limitation of the study)
Author Response
Open Review
(x) I would not like to sign my review report
( ) I would like to sign my review report
English language and style
(x) Extensive editing of English language and style required
( ) Moderate English changes required
( ) English language and style are fine/minor spell check required
( ) I don't feel qualified to judge about the English language and style
Comments and Suggestions for Authors
The authors describe viral parameters of cats naturally infected with FIV or FeLV and treated orally with low doses of human interferon alpha. The aim of the study in the abstract indicates that also clinical parameters are investigated; however, these results are not presented in detail. The manuscript presents the viral parameters of FeLV-infected cats and proviral loads of FIV-infected cats under therapy (single-arm study). The results are interesting: a decrease of the FeLV capsid antigen was found, followed by a rebound effect after the therapy was stopped. These data should be published. However, the manuscript needs major revision. Some sections are difficult to understand, possibly due to language issues. The results should be streamlined, and statistics revised. The clinical data should be added to the manuscript to present a complete story.
We thank the reviewer for all his/her comments, most of which have been incorporated in the manuscript.
Title and throughout the text: “evolution” implies – at least in biological terms – that the virus changed the characteristics. This was not investigated. Please delete or replace this term.
Evolution has been replaced by Progression all the times it appears.
Abstract: the authors should substantiate their statements by adding numbers, percentages, significances. Examples:
- line 17: “in vitro … shedding of infectious particles” – unclear what is meant here. Virus is usually shed by animals; thus, in vivo. Please rephrase, since here in concerns in vitro. Shedding has been replaced by release.
- line 20: “belonging to households” – privately owned cats? The suggestion of the reviewer has been incorporated.
- line 25: “most animals”. Can the authors give a number or a percentage? We have included the number in the abstract (15 of the 16 FeLV+ symptomatic cats; 20 of the 22 FIV+ symptomatic cats).
- line 26: was this a significant decrease? No, it was not statistically significant.
Introduction
- Language could be improved. E.g. line 37 “long-life persistent”; line 19: FeLV is not a disease; it’s a virus. Sorry for the mistake. It has been changed to lifelong persistent. The sentence has been slightly rephrased in order to indicate that what it was meant was feline leukemia and feline immunodeficiency pathogenesis.
- FeLV pathogenesis: The authors do not use the currently used terminology for the different courses of FeLV infection; this needs to be adapted to the recent literature. I suggest using the following terms: “progressive infection” and “regressive infection”. The paragraph has been rephrased.
Line 53/53: a cat that is progressively infected does not have latent infection. This sentence is contradictory. Line 54: during which infection? Probably progressive infection is meant here. The paragraph has been rephrased.
Line 58: “persistent viremia”, probably progressive infection is meant. Line 60/61: probably progressive infection is meant here? Is reference 4 the right reference for this statement (lifespan 1-3 years in cats with progressive infection)? The paragraph has been rephrased and the reference changed.
Aim of the study: How is this aim is different from that of previous studies and how does it add new information to the field? Also see: “the use of interferon as antiretroviral therapy in cats has been extensively studied” (line 79). The sentence has been rephrased.
Lines 86-87: High doses of human interferon-alpha induced anti-IFN antibodies. Could the authors please add a reference to this sentence? Done.
Lines 89/90: I am not quite convinced about this general statement that human interferon alpha has advantages over the feline interferon omega. E.g. compare the FeLV guideline by ABCD, table 3, summary of immunomodulatory substances against FeLV. We are sorry to disagree with the reviewer. We think that rHuIFN-α has over rFeIFN-ω at least the advantages of easy administration, being inexpensive, and in our hands has produced a sustained clinical improvement.
M&M
Lines 146-148. Could the authors please list “the clinical signs most frequently observed in feline retrovirosis”? Did they use a clinical score sheet? If so, please add to the publication. Please delete the term “feline retrovirosis”. We have listed some of the clinical signs most frequently observed. The complete list is in the reference provided (Collado et al., 2012), where the clinical score sheet is also presented. We have substituted feline retrovirosis by FeLV- and FIV-infections. However, we were not aware that feline retrovirosis was an incorrect term to use.
Please do NOT use rtPCR for real-time PCR. This is confusing, since RT-PCR is used for reverse transcriptase PCR. Please list this parameter (real-time PCR result) in the results section as “proviral load”. Use “RT activity” instead of only “RT” in the results section (for RT activity). Done.
Why did the authors used heparinized blood for PCR? Heparin can inhibit polymerase activity. EDTA or sodium citrate would be more adequate. Maybe add to limitations of study in the discussion? When we collected blood from the cats we did it in two tubes, one with EDTA and the other with heparin. We had used the EDTA tube for the hemogram and CD4/CD8 ratio and the heparin one for the remaining analysis, either in the plasma (viral proteins, RT activity, antibodies against rHuIFN) or extracting DNA from the cells. We thought that as we had extracted the DNA from the blood with a DNeasy Tissue Kit, heparin would be absolutely removed and would not interfere. However, as suggested by the reviewer, we have mentioned that real-time PCR was done from heparinized blood in a paragraph about limitations of the study in the discussion section.
Why did the authors use a human specific GAPDH assay? There are feline assays published. Did the authors make a sequence comparison or is there published data on the specificity of the assay for feline genomic DNA? Maybe add to limitations of study in the discussion? We have always used GAPDH with good results. Gene gapdh is well conserved between mammals. Initially we had performed a sequence comparison (feline gapdh NM_001009307; human gapdh NM_001357943) and we had detected no problem in silico. We have not added it to limitations of the study.
Data analyses: Line 186: why did the authors use parametric tests (ANOVA) – was the data normally distributed? Line 199: statistics: “some data are missing”. It should be specified, which data was missing/available for statistical analysis. A sentence has been included. The fact that data were missing have also been discussed in the limitations of the study.
According to the test of Shapiro-Wilk, real-time PCR has a normal distribution, while the CS, p27 and RT activity do not. To determine whether the differences are enough to support that interferon improved the clinical and viral status of treated cats, average values were compared using T-Student for associated samples for real-time PCR, while for the CS, p27 and RT activity we applied Wilcoxon test.
Result
Lines 190 ff: the clinical scores cannot be judged like that. The conclusion that during the study period most of the cats improved clinically is not supported by data and statistics. Please add detailed results. We have included results and statistics.
Figure 1: the content of the circles, particularly the colored sections, is hard to read. There seems to be a mislabeling (twice FeLV RT). The type of graph has been changed to stacked columns, where we believe that the numbers can be read better.
Figure 2: only present the mg/ml in Fig. 2; consider using box plot or similar, since probably the data is not normally distributed. The percentages are suggestive of a larger change than present; delete from figure (may be kept in the text). We believe that by doing this, some data would be lost. As other reviewer had asked to present more data in the same way as Figure 2, we have not changed it.
Minor: The tables should be mentioned in the text in the correct order. Now Table 2 is mentioned in line 201, then table 1 in line 231. Done.
RT results and provirus results: I suggest presenting these results also in a figure (like Fig. 2), where the reader can assess the actual data at each time point and judge by him/herself the de-/increase of the activity over time. Table 1 seems not helpful. Done. Table 1 has been deleted.
Provirus results (line 250): is there any justification to consider a change of 20% to be biological relevant? It was an arbitrary value that we considered would be enough to eliminate results happening by chance.
Please delete chapter 2.3. and table 2. Redundant. Sections have been renamed.
Discussion
Why not start with the discussion of the most relevant results? Transient decrease in FeLV viral loads (lines 312 ff). Please discuss, why the decrease was only transient. Could the therapy be continued or restarted? What about development of anti-human IFN alpha antibodies? Could this be measured (at least discuss)? Done.
Move the low toxicity up (as a positive statement; lines 376 ff), plus low-cost, plus p.o. administration. We do not understand the suggestion. However, we have modified the paragraph and hope it is more understandable.
Move FIV detection further down (since mainly negative). Done
Line 316 – sentence does not make sense. FeLV p27 IS circulating antigen (“is a measure of circulating antigen”?) We have rephrased the sentence.
Please do not mix FIV and FeLV in the discussion section. Done.
Lines 366 ff: not sure I understand: “upon the discontinuation of treatment, FeLV-positive cats could develop persistent viremia”. Did you not include only cats with progressive infection (persistent viremia)? If this was not the case, please add to discussion (also cats with regressive infection could have been included, since the cats were tested only at one timepoint for antigenemia prior to the start of the study? Clear limitation of the study). Only cats with progressive infection were included. We tested antigenemia (p27CA) at each visit.
We again thank the reviewer for his/her valuable comments.
Submission Date
05 July 2019
Date of this review
06 Aug 2019 10:20:54
Reviewer 3 Report
Evolution of the viral parameters in FeLV- or FIV-naturally infected cats treated orally with low doses of human interferon alpha.
Summary:
This manuscript by Gomez-Lucia et al describes a clinical study in which human interferon alpha was used to treat naturally FeLV- and FIV-infected cats. Due to the lack of effective antiviral therapeutics for treatment of feline retroviral induced diseases it is important that these studies are undertaken. An interesting outcome of the study was the improvement of the clinical score in FeLV positive treated cats during the observation period. Unfortunately, there wasn’t a sustained decrease in viral p27antigenemia and reverse transcriptase activity indicating decreased virus production after treatment was discontinued. Their study also shows that human interferon alpha treatment was not as effective in FIV-infected cats. The data presented in this paper represents the average of all the cats evaluated at each time point. The authors state in the Discussion that “..average values are not easy to use as they do not provide a true measure of the evolution in each cat”. An evaluation of the longitudinal data on individual cats might elucidate differences with respect to treatment efficacy.
Specific Comments:
Title: The term “evolution” is not the correct term to be used to describe the data presented in this manuscript. A more appropriate term would be “evaluation” of the viral parameters. The term evolution is used in several places throughout the manuscript and should be changed. Page 5, lines 198-201: This section of the manuscript indicates that data is missing from some of the cats initially enrolled in the study. Table 2 provides data on the clinical status of the cats from this study. There is no indication of the number of cats from which there is a complete data set. The data presented in Table 2 provides the average of the clinical score, concentration of FeLV-p27, RT activity and viral load. The standard deviation or SEM should be included in the table. How many cats improved their clinical situation? Viral load should be changed to Proviral load. Page 5, line 201 and page 9, lines 289-292: There is reference to a manuscript in preparation. How is the data presented in the manuscript currently under review different from that being proposed for manuscript in preparation? Page 1, line 27, page 5, line 209, page 7, line 238, Page 10, line340, page 11, line 395: Provide the parameters/criteria used to define the “rebound effect/pattern”. Page 5, line 211: typo-en should be in. The units for the range of p27CA should be included (0.46-0.60 mg/mL). Page 7, line 228: This section should be labeled 3.2 Page 7, line 231: Instead of stating “around three fourths of the cats” provide the actual number of cats with detectable RT activity at M0 with reduced RT activity at M2, M4 and M10. Page 8, line 247: This section should be labeled 3.3 Page 8, line 256-257: The term parallel evolution does not accurately represent the data. Perhaps what is meant is that there is a correlation between the concentration of FeLV-p27 and proviral load, which is indicated by the inclusion of a statistically significant p value. Page 8, line 267: This section should be labeled 3.4 Page8, line 269: The data presented in this manuscript is not adequate to evaluate viral evolution. One would need to identify the genetic changes in the genome of the viruses to address viral evolution. Page10, line 333: evolution is not the appropriate term- perhaps what is meant is progression of the infection Discussion: Treatment of cats with human interferon alpha is known to induce production of antibodies against hIFN-alpha. While not evaluated in the current study this should be addressed in the discussion.Author Response
Please, see the attachment.

Round 2
Reviewer 1 Report
The authors have addressed satisfactory all of my concerns.
Reviewer 2 Report
The manuscript has been significantly improved by the authors and is in my opinion acceptable for publication in this form.
Reviewer 3 Report
the authors addressed the reviewers comments